# Influenza and Universal Vaccine Research in China

**DOI:** 10.3390/v15010116

**Published:** 2022-12-30

**Authors:** Jiali Li, Yifan Zhang, Xinglong Zhang, Longding Liu

**Affiliations:** Key Laboratory of Systemic Innovative Research on Virus Vaccine, Institute of Medical Biology, Chinese Academy of Medical Sciences and Peking Union Medical College, Kunming 650118, China

**Keywords:** influenza, influenza virus, influenza vaccine, vaccine development

## Abstract

Influenza viruses usually cause seasonal influenza epidemics and influenza pandemics, resulting in acute respiratory illness and, in severe cases, multiple organ complications and even death, posing a serious global and human health burden. Compared with other countries, China has a large population base and a large number of influenza cases and deaths. Currently, influenza vaccination remains the most cost-effective and efficient way to prevent and control influenza, which can significantly reduce the risk of influenza virus infection and serious complications. The antigenicity of the influenza vaccine exhibits good protective efficacy when matched to the seasonal epidemic strain. However, when influenza viruses undergo rapid and sustained antigenic drift resulting in a mismatch between the vaccine strain and the epidemic strain, the protective effect is greatly reduced. As a result, the flu vaccine must be reformulated and readministered annually, causing a significant drain on human and financial resources. Therefore, the development of a universal influenza vaccine is necessary for the complete fight against the influenza virus. By statistically analyzing cases related to influenza virus infection and death in China in recent years, this paper describes the existing marketed vaccines, vaccine distribution and vaccination in China and summarizes the candidate immunogens designed based on the structure of influenza virus, hoping to provide ideas for the design and development of new influenza vaccines in the future.

## 1. Introduction

Influenza is an acute respiratory infection caused by the influenza virus, which seriously endangers human and animal health and results in death in severe cases [1]. Since the 20th century, there have been several influenza pandemics in human history, the most catastrophic being the “Spanish Flu” in 1918–1919, which killed about 50 million people worldwide [2], as well as the “Asian Flu” in 1957–1958, the “Hong Kong Flu” in 1968–1969, the “Swine Flu” in 1976, the “Russian Flu” in 1977, and the Influenza A in 2009, which also caused millions of deaths [3]. On the other hand, human infections with avian influenza viruses (AIV) represent a persistent public health threat. AIV are divided into two pathotypes, low pathogenic avian influenza (LPAI) and highly pathogenic avian influenza (HPAI), based on their virulence in chicken. The first recorded human deaths caused by HPAI virus occurred in Hong Kong in 1997 when an H5N1 virus infected 18 individuals with six fatalities. The second major AIV zoonosis was caused by an H7N9 virus, which was first detected in China in 2013. It has since caused more than 1500 human cases at a case fatality rate of 39% [4]. The control of avian influenza has become a common and daunting task for human public health [5,6,7]. Current influenza virus vaccines are effective, but only if they are well-matched to the prevalent strain. Pandemic influenza and seasonal influenza cause considerable morbidity and mortality worldwide, posing a serious threat to public health and the global economy, and the World Health Organization (WHO) estimates that seasonal influenza causes approximately 3–5 million severe cases and 290,000–650,000 deaths annually. Pregnant women, infants and young children, elderly individuals and patients with chronic underlying diseases are at higher risks of serious illness and death from influenza [8]. With a population of over 1.4 billion, China has a large burden of influenza morbidity and mortality, but the development of the influenza epidemic is not clear enough. In this paper, we statistically analyze the cases related to influenza virus infections and deaths in China in recent years, describe the existing marketed vaccines, vaccine distribution and vaccination in China, and also summarize multiple influenza vaccine candidate immunogens and vaccine design strategies to provide ideas for the design and development of new influenza vaccines in the future.

## 2. Epidemiology of Influenza in China

Influenza virus research in China began in 1952 and led to the establishment of the National Influenza Center in 1957, followed by the establishment of an influenza surveillance network in 2000, focusing on influenza-like case reporting and virus isolation, which has comprehensively improved the overall capacity of influenza surveillance in China. Since 2009, China’s influenza surveillance network has been gradually expanded and improved and now includes 554 sentinel hospitals and 410 network surveillance laboratories covering all provinces, municipalities and autonomous regions in mainland China. Sentinel hospitals report cases of influenza-like illness (ILI) to the China National Influenza Surveillance Information System (CNISIS) and collect respiratory samples. The network laboratories use real-time reverse transcription polymerase chain reaction (RT-PCR) to determine whether the collected samples are positive for influenza virus [9].

### 2.1. Analysis of Influenza Infection and Death Cases in China in Recent Years

With a total population of more than 1.4 billion, the burden of influenza morbidity and mortality in mainland China is high. According to the data on influenza infections and deaths in mainland China from March 2009 to August 2022 (as of 17 November 2022, Figure 1a,b) reported by the Bureau of Disease Prevention and Control of the National Health and Health Commission, the influenza epidemics and influenza-related deaths were concentrated mainly in the winter season (January–March and November–December), and the incidence of influenza was reported to be the lowest in the July–October period. The 2019 influenza epidemic was severe and longest-lasting, in February-June as well as November-December, with the highest number of influenza cases in December, about 1.2 million people (Figure 1a). The following January had the highest number of deaths at 108. It was reported that influenza-related deaths were concentrated in the January–March and November–December periods (Figure 1b), but the influenza case fatality rate showed no seasonal pattern (Figure 1c). 

During 2009–2017, the number of influenza infections and deaths per year was relatively low, with low morbidity and mortality (Figure 1d,e), but during 2018–2020, the number of influenza infections and deaths increased substantially, as did the mortality rate, probably due to the expansion of the influenza surveillance network and more timely and accurate reporting of influenza cases. The number of seasonal influenza infections and associated deaths after 2019 were lower than before the emergence of SARS-CoV-2, probably because preventive control of SARS-CoV-2 somewhat also controlled the influenza epidemic. The overall annual morbidity and mortality rates were not clearly characterized (Figure 1f). Subsequently, the number of influenza cases was low in 2021, except for a mild epidemic during November–December.

### 2.2. Influenza Virus Subtypes and Seasonal and Regional Characteristics in China

Currently, the main influenza virus subtypes causing seasonal influenza in China are influenza A/H1N1/pdm09, influenza A/H3N2 and influenza B viruses. According to a review and analysis of influenza-related health outcomes in mainland China by Diamond et al., each influenza subtype exhibited significantly different mean epidemic durations, with the influenza A H3N2 epidemic having the longest mean duration of approximately 4.83 months/year, followed by influenza B and influenza A/H1N1pdm09 with mean lengths of 3.57 and 2.33 months/year, respectively. Independent of latitude, influenza A/H1N1/pdm09 and B virus spectra occurred mainly in the winter, peaking in January and February, respectively, while influenza A/H3N2 viruses varied widely across the country, with winter epidemics in high latitudes (e.g., Liaoning Province) and summer epidemics in mid- and low-latitude regions (e.g., Fujian Province). Among all provinces, January was the peak month of activity, with an average positivity rate of 21.8%, and October had the lowest average positivity rate of 4.37%. In addition, the annual influenza epidemic in mainland China varies according to latitude. Influenza epidemics in high latitude provinces in China are characterized by epidemics in the winter with short durations but strong epidemic intensity, while provinces at middle and low latitudes have weak epidemics but long durations as semiannual epidemics (e.g., Zhejiang and Anhui provinces) or year-round epidemics (e.g., Hainan province) [10].

### 2.3. Influenza Epidemic Causes Disease and Economic Burden

According to ILI surveillance and virological data from national sentinel hospitals, Feng et al., calculated the national average rate of influenza-related consultations in 30 Chinese provinces from 2006 to 2015 and found that there are 2.5 consultations per 1000 people, and this peaked at 7.8 consultations per 1000 people during the 2009 influenza pandemic [11]. Gong et al., performed a literature review and estimated the average annual number of excess outpatient emergency department visits for ILI due to influenza in China from 2006–2019 to be 2,346,000 to 3,005,000 hospitalizations for severe acute respiratory infections and 92,000 excess deaths from respiratory diseases [12]; Li et al., fitted a linear regression model with data from 22 of 31 Chinese provinces and estimated that, from 2010–2015, an average of 88,100 influenza-associated excess respiratory deaths occurred per year in China during that period, equivalent to 8.2% of all respiratory deaths [13]. Notably, influenza-associated excess respiratory mortality valuations exhibited spatial variation across the provinces studied, i.e., valuations in eastern and western provinces of China were higher than those estimated for central provinces, possibly due to incomplete health care services in the west and high population density in the east, which increases the risk of influenza transmission [13].

In addition to the severe burden on people’s lives and health, influenza causes huge economic losses to individuals and countries, with indirect economic losses reaching tens of billions of dollars globally each year [14]. China is an influenza-prone region, and most studies on the economic burden associated with influenza have focused on more economically developed regions; for example, Beijing, Shanghai and Tianjin have the highest ILI burden, and Qinghai, Gansu and Ningxia have the lowest ILI burden. This may be related to differences in economic development and different health-seeking behavioral patterns between provinces, where individuals may choose medication rather than consulting an outpatient clinic in provinces with average economic development [11]. Direct medical costs for outpatient influenza cases in China range from CNY 186 to 804; direct nonmedical costs range from CNY 7 to 212; indirect costs range from CNY 139 to 963; and total economic burden ranges from CNY 464 to 1320. Furthermore, direct medical costs for inpatient influenza cases range from CNY 2625 to 20,712; direct nonmedical costs range from CNY 1200 to 1809; and indirect costs range from CNY 204 to 2408. The total economic burden was in the range of CNY 9832–25,768. Chen et al., assessed the annual economic loss caused by seasonal influenza in China from 2011–2019 as approximately CNY 33 billion to CNY 106 billion, which is equivalent to 0.03% to 0.1% of China’s GDP in 2019. In 2019 alone, the total economic burden associated with influenza in the country was CNY 26.381 billion, accounting for approximately 0.266% of the GDP in that year, with hospitalized cases, emergency outpatient cases and premature deaths causing 86.4%, 11.3% and 2.4% of the total economic burden, respectively [12].

The above studies illustrate that influenza epidemics not only cause serious health effects but also impose a significant economic burden, that measures to control influenza epidemics are urgently needed and that vaccination is the most cost-effective means to do so.

## 3. Vaccines and Vaccination Status in China

Influenza vaccination is an effective means of preventing and reducing influenza-associated severe illness and death, as well as reducing the health risks associated with influenza-associated illness and the use of health care resources [15]. The trivalent vaccine includes one of the influenza A H1N1, A H3N2 and influenza B virus Victoria or Yamagata strains, while the quadrivalent vaccine includes influenza A H1N1, H3N2 and influenza B virus Victoria and Yamagata strains, covering a wide range. The WHO recommended trivalent influenza vaccine components for 2022–2023 based on chicken embryos produced in the Northern Hemisphere: A/Victoria/2570/2019 (H1N1) pdm09 analog, A/Darwin/9/2021 (H3N2) analog and B/Austria/1359417/2021 (Victoria lineage) analogs. The quadrivalent influenza vaccine component contains two strains of the B strain, the three abovementioned strains and the B/Phuket/3073/2013 (Yamagata lineage) analog. Compared to the previous year, the A (H3N2) subtype and B (Victoria) lineage viruses changed strains [16].

The influenza vaccines now approved for marketing in China are trivalent inactivated influenza vaccine (IIV3), quadrivalent inactivated influenza vaccine (IIV4) and trivalent live attenuated vaccine (LAIV3). IIV3 includes both inactivated and subunit vaccines; IIV4 is an inactivated vaccine; and LAIV3 is an attenuated vaccine. As of 14 August 2022, 11 manufacturers have supplied influenza vaccine, and information on specific influenza vaccine manufacturers and their products is shown in Table 1.

Influenza continues to pose a significant threat to public health, and vaccination is currently the most effective response to influenza. Currently, the effectiveness of polyvalent influenza vaccines can be improved by adding adjuvants to stimulate immunogenicity, optimizing means of virus strain prediction and using high doses of injectable vaccine for immunocompromised populations.

The inactivated influenza virus vaccine for infants aged 6–35 months is 0.25 mL/dose, containing 7.5 micrograms of each HA antigenic component, and generally requires two doses to achieve immune protection; the inactivated influenza virus vaccine for children aged 3 years and older is 0.5 mL/dose, containing 15 micrograms of each HA antigenic component, and requires only one dose. The standard for the antigenic content of the trivalent live attenuated vaccine is an attenuated virus titer of not less than 6.9 lgEID50 (50% egg infectious dose, EID50) for each of the two subtypes of influenza A virus H3N2 and H1N1 and an attenuated virus titer of not less than 6.4 lgEID50 for the Victoria lineage of influenza B virus, with a size of 0.2 mL/dose, for 3- to 17-year-old children and adolescents aged 3–17 years [16].

The statistics of the batch issue volume of China’s influenza vaccine industry from 2015 to 2020 are shown in Figure 2. The batch issue volume of China’s influenza vaccine in 2018 was 16,123,900 doses, which was the lowest in recent years. This increased in 2019, with an influenza vaccine batch issue volume of 30,784,200, 14,660,300 doses more than the previous year. The batch issue volume of the influenza vaccine in 2020 reached 57,519,900 doses, an increase of 26,735,700 doses over the previous year, and the number of approved influenza vaccines will not be announced after 2021 [9]. China’s quadrivalent influenza vaccine was first approved for marketing in 2018, and according to statistics, 5,122,500, 9,710,500 and 33,582,300 doses were approved during 2018, 2019 and 2020, respectively. The number of quadrivalent influenza vaccines issued has increased year by year since its launch, with the proportion exceeding that of trivalent influenza vaccines in 2020, reaching 58.38%.

According to statistics, the average vaccination rate of influenza vaccine in China was 2.43% from 2014 to 2020, with the lowest rate of 1% in 2018 and 4% by 2020, while according to the Centers for Disease Control and Prevention, 175 million doses of influenza vaccines were distributed in the United States during the 2020 influenza season, with an adult vaccination rate of approximately 48.4%, compared to less than 1 in 10 in China [17]. Possible reasons for this low vaccination rate are that China has not yet included the influenza vaccine in its national immunization program, limited coverage of areas where the free vaccination policy is implemented, insufficient awareness of influenza and influenza vaccines among the population, insufficient vaccine supply, single variety of existing influenza vaccine and the need for annual revaccination [18,19,20]. Due to the current low influenza vaccination rate in China and the fragile protection of the herd immune barrier, to a certain extent, influenza epidemics can easily develop into epidemic trends. The influenza vaccination rate in China still needs to be further improved, and measures can be taken to: first, gradually incorporate influenza vaccination for specific risk groups into national immunization planning programs and local public health programs. Second, vaccination should be increased for target groups, such as children, elderly individuals and medical personnel. Third, efforts should promote combined vaccinations and actively optimize the combined vaccination procedures for influenza and other vaccines as a way to improve the efficiency of immunization services [21].

## 4. Development of Universal Influenza Vaccine

The influenza virus, belonging to the Orthomyxoviridae family, is an enveloped single-stranded negative-sense RNA virus whose genome encodes 11 proteins, namely RNA polymerase subunit proteins (PB1, PB2 and PA), surface glycoprotein hemagglutinin (HA), neuraminidase (NA), nucleoprotein (Nucleoprotein, NP), matrix protein (M1), membrane protein (M2) and nonstructural proteins NS1, NS2 and PB1-F2 [22], which play important roles in influenza virus replication (Figure 3). According to the NP and M1 antigenic determinant clusters, influenza viruses can be classified into four types: influenza A virus (IAV), B virus (IBV), C virus (ICV) and D virus (IDV) [23]. Human influenza is mainly caused by influenza A and B viruses, and there are 18 HA subtypes and 11 NA subtypes, according to HA and NA classification system. Influenza B viruses are mainly divided into two lineages, the Victoria lineage and Yamagata lineage.

Influenza viruses mutate mainly through antigenic drift and antigenic transformation to evade natural infection and vaccination-induced herd immunity [24]. Antigenic drift occurs when an error in the virus replication process leads to a change in the antigenic epitope, resulting in a new viral subtype [8], while antigenic transformation occurs less frequently when two or more influenza viruses infect the same cell, and the interchange of viral gene fragments leads to the generation of new subtype combinations in daughter viruses. Mutation of the virus leads to a change in the conformation of the neutralizing antibody recognition epitope, allowing it to easily evade the specific immunity induced by current vaccines, triggering annual seasonal influenza epidemics and global influenza pandemics [25,26,27,28].

The existing influenza vaccination protection rate is only 40–60%; influenza viruses are prone to mutation; and the effectiveness of the vaccine is further reduced when new influenza virus strains emerge that do not match the vaccine strain [29,30]. Therefore, the annual influenza virus vaccine needs to be updated and then readministered to the population based on new virus strains in order to be effective at preventing influenza. The production process of influenza vaccines takes at least 6 months, and most of the currently licensed vaccines are produced in chicken embryos, which sometimes leads to adaptive mutations of the virus, thus reducing vaccine efficacy [31]. Therefore, the development of a universal influenza vaccine is important to prevent multiple influenza virus strains and the global pandemics they cause. Using highly conserved viral protein regions among the viral subtypes as potential target antigens is currently the main way to develop and design a universal influenza vaccine, and the main targets include HA, NA, M2e and NP.

### 4.1. Hemagglutinin (HA) as an Immunogen for Universal Vaccine Design

Hemagglutinin (HA) is the most abundant glycoprotein on the surface of influenza viruses and plays a key role in the process of viral invasion and infection [32]. In terms of spatial conformation, each HA monomer consists of two parts: the globular head consists of HA1, which allows the virus to adsorb to the host cell by binding to the sialic acid receptor, and the rod-like stem, which consists of HA2 and part of HA1, and is anchored in the lipid bilayer of the viral capsid. After the HA head binds to the cell surface sialic acid, the stem mediates the fusion of the viral capsid and host cell membrane and the viral release of the capsid. Since the HA head contains antigenic determinant clusters and receptor binding sites, it is an important target antigen for vaccine development [33], and current seasonal vaccines also mainly trigger antibodies against the structural domain of the HA head; however, this site is prone to antigenic drift and transformation, resulting in limited vaccine protection. The highly conserved HA stem, which produces protective antibodies that neutralize viral infectivity without inhibiting hemagglutination, is a more attractive target for the development of a universal influenza vaccine. Many studies have shown that this site contains many antigenic epitopes that bind broad-spectrum neutralizing antibodies (bnAbs) and has an important role in inhibiting viral entry into cells [34,35,36,37].

The HA2 region is a potential research target for a universal influenza vaccine [38], but since the antigenic epitopes of the HA stem are often blocked by its head, vaccine design needs to increase the exposure of the HA stem to the host immune system, such as by using certain short peptide epitopes in the structural domain of the stem to bind to carrier proteins to increase the immune exposure of the HA stem and help antigen presentation into the structural domain of the stem, thus enhancing the immune effect. Current vaccine development strategies based on influenza virus HA can be divided into headless HA vaccines, chimeric HA vaccines, mosaic HA vaccines and computationally optimized broadly reactive antigen (COBRA) HA vaccines [39,40,41,42] (Figure 4).

#### 4.1.1. Headless HA Vaccine

One such strategy to increase the immunological advantage of HA stems is to completely remove the HA head region, creating a “headless” HA structure. Headless HA is difficult to develop because it is not stable, and most cross-reactive HA stem antibodies have conformation-dependent epitopes. For example, Wohlbold et al., developed an antigen based on HA stems that induced protection against influenza strains but was not recognized by any antibodies that neutralized the stems due to the lack of the correct conformation of the natural stem structural domain [43]. To address the dependence of HA stem epitopes on the natural conformation, subsequent studies enhanced the structural integrity of stem immunogens by rational design to mimic the natural trimeric conformation. For example, Corbett et al., constructed HA stem trimer self-assembled nanoparticles with an intact structure and demonstrated protection against lethal homozygous influenza virus attack after immunization of mice [44]; Darricarrère et al., presented headless HA stem antigen on ferritin nanoparticles coupled with AF03 adjuvant delivery and produced antibodies in nonhuman primates that neutralized multiple influenza virus strains, indicating that these vaccines can provide broad protection against influenza infection in vivo [45].

#### 4.1.2. Chimeric HA Vaccine

Another strategy to induce high levels of anti-stem HA antibodies is sequential immunization with chimeric HA molecules (CHAs), which contain the same HA stems and different HA heads, allowing the induced antibody response to focus primarily on immunizing against the subdominant HA stems. To date, several influenza vaccine platforms have successfully induced HA stem-reactive antibodies in animal models using chimeric HA molecules. Liao et al., demonstrated that a monoglycosylated CHA vaccine with shared H5 as the bulbous head and shared H1 as the stem induced more stem-specific antibodies, had a higher ADCC, better neutralized the viral subtypes under study and had broad protective activity against heterologous influenza viruses (including H1, H3, H5 and H7 viruses) [46]. Kotey et al., designed two CHAs consisting mainly of the conserved region of H1: H1/H5 HA and H1/H9 HA, which immunized all mice from infectious attack by H1 virus, and the sera of immunized mice showed cross-reactivity with H5 subtype strains [47]. In addition, there are CHAs that have entered clinical trials, which consist of the head of H8 and H5 subtypes with the stem of H1, which induced a broad, strong and durable immune response after human inoculations [38]. Therefore, based on the above findings, CHA has the potential to be developed as a universal vaccine.

#### 4.1.3. Mosaic HA Vaccine

Rather than altering the entire head structural domain of HA to produce cHA, these immunodominant antigenic sites in the head should be replaced with sequences of other HA isoforms to produce broader protective potency. For example, Sun et al., replaced the major antigenic site of influenza B virus HA with the corresponding sequence of influenza A virus HA to generate mosaic HA expressed as a soluble trimeric protein and showed that the vaccine was cross-protective against homologous and heterologous influenza B virus strains in a mouse model [41]; Corder et al., designed a new mosaic H1 HA immunogen involving multiple IAVs, which after immunization of mice, showed induction of a strong antibody response and cellular immunity, exhibiting strong cross-protection against different IAV strains [48]; Liu et al., produced two inactivated influenza B virus mosaic HA vaccines (showing homologous or heterologous HA structures) and showed that both methods induced a long-lasting and cross-protective antibody response, showing strong ADCC activity [49], further demonstrating the potential of mosaic vaccines as general-purpose influenza vaccines.

#### 4.1.4. COBRA HA Vaccine

The most widely studied optimized HA protein, called computationally optimized broadly reactive antigen (COBRA), can be used as a recombinant protein or displayed on the surface of viruses or VLPs and has been shown to induce multiple antigenic variants in a variety of animal models, including mice, ferrets and nonhuman primates, with the most widely hemagglutination-inhibiting responses to serum. Reneer et al., constructed an H2 COBRA HA vaccine and tested it in ferrets immunized against H2N3 attack, resulting in vaccinated ferrets exhibiting higher antibody titers and recognizing the highest number of H2 influenza viruses in a classical neutralization assay compared to wild-type H2 HA recombinant proteins [50]; Huang et al., used the COBRA approach to generate two new candidate COBRA HA vaccines, Y2 and Y4, from H1N1 isolates prevalent during 2013–2019. Mice vaccinated with Y2 and Y4 effectively reduced morbidity and mortality after infection with H1N1 influenza viruses, and vaccine-initiated antibodies associated with H1N1 isolates from 2009 to 2021 and pdm09-like viruses produced a high degree of cross-reactivity, with the strongest intensity of cross-reactivity in 2019–2021 [51]; Allen et al., first infected ferrets with three historical H3N2 species and then tested for expanded immune breadth with constructed COBRA or wild-type (WT) H3 VLP vaccine antigens and showed that, compared to the WT group, the COBRA group in all three historical H3N2 strains produced more cross-reactive antibodies and induced antibodies capable of neutralizing live virus infections of modern drifting H3N2 strains at higher titers [52].

### 4.2. Neuraminidase (NA) as a Candidate Immunogen for Broad Protection

Neuraminidase, another major glycoprotein on the surface of influenza viruses, plays an important role in virus transmission by cleaving the viral sialic acid receptor on the host cell membrane to facilitate the release of viral particles and prevent the accumulation of offspring viruses on the cell surface [53,54]. It is usually composed of four identical monomers noncovalently bound to form a mushroom-like tetramer, a structure that helps maintain the stability of its activity [54]. NA can be divided into four major regions from the N-terminal to the C-terminal ends: the cytoplasmic, transmembrane, stem and head regions. The cytoplasmic region is highly conserved in sequence and is associated with virus assembly and outgrowth [55]; the transmembrane region is associated with virus outgrowth; the stem region is highly variable and poorly conserved in sequence, and its length affects neuraminidase enzyme activity [56]; and the NA head region contains the NA protein antigenic determinant cluster, the neuraminidase activity center and the glycosylation site [57]. NA proteins induce the body to produce NA-specific antibodies, which play an important role in inhibiting virus spread and controlling influenza infection through antibody-dependent cell-mediated cytotoxicity (ADCC) and complement-dependent cytotoxicity (CDC) clearance [58], and it has been demonstrated that NA has a lower antigenic drift frequency than HA, is relatively genetically conserved, can bind to antigenically conserved epitopes in the same NA subtype and may provide a broader protective range, making it an ideal antigen for a universal influenza vaccine [59].

Kim et al., constructed an NA VLP vaccine against influenza virus H1N1 pdm09, which induced virus-specific antibody responses after immunization of mice and generated cross-protective immunity against different influenza viruses and further found that sera from mice immunized with this vaccine were cross-protective in young mice [60]. Strohmeier et al., used a tetramerization of the phosphoprotein from the measles virus and replaced the cytoplasmic region, transmembrane region and stem region of NA with a tetramerized structural domain from measles virus phosphoprotein to design a recombinant vaccine, N1-MPP. This vaccine retained the antigenicity, activity and structural integrity of NA and provided strong protection against lethal viral attack in mice, resulting in a diminished NA response when mixed with a seasonal vaccine of a different formulation, but induced a strong antibody response when given separately; thus, this recombinant NA vaccine may enhance and amplify protection against seasonal influenza viruses [61]. Additionally, Skarlupka et al., produced N1 vaccine antigens containing NA activity of four influenza A viruses using the COBRA method, and when attacked with the four influenza viruses in question, immunized mice were protected from death and had low lung virus titers with little change in weight loss. Thus, the vaccine antigens may become complementary components of a multiantigen universal influenza virus vaccine formulation that also contains HA antigens, and it is hypothesized that its use in combination with HA antigens could extend protection against various virus strains [62].

### 4.3. Membrane Protein (M2) Extracellular Region (M2e) as a Candidate Universal Immunogen for Vaccine Design

Membrane protein (M2) is a transmembrane protein consisting of 96 amino acids divided into extracellular, transmembrane and cytoplasmic regions, and is usually present as a tetramer on the surface of the influenza virus vesicle membrane, forming a proton-selective ion channel regulated by pH, which is activated in an acidic environment at low pH, allowing the proton to be pumped into the viral particle, participating in the processes of viral exfoliation. The extracellular region of the M2 protein, known as M2e, contains only 24 amino acid residues and is exposed on the surface of the viral capsid or on the surface of infected cells and is associated with the assembly of nascent viral particles. Because M2e is highly conserved across different strains and subtypes of influenza viruses, it can induce a cross-protective immune response, making it an attractive target during the development of a universal influenza vaccine. Previous experiments have demonstrated that anti-M2e antibodies bind strongly to M2e bound at the host cell surface and inhibit viral replication and outgrowth, thus producing protection in animals. Due to the weak immunogenicity of M2e itself, many strategies have been used to enhance the immunogenicity of M2e-based vaccines [63]. 

For example, by increasing the density of M2e epitopes in vaccines to significantly improve immunogenicity and efficacy, Krishnavajhala et al., expressed M2e peptides at high density on Sindbis virus and immunized mice with M2e-specific antibodies that protected them from lethal virulence attack of influenza A virus [64]. Furthermore, they self-assembled multiple tandem repeats of M2e into M2e VLP, which was then incorporated into the polymer delivery system AS04 and subcutaneously immunized mice showing immunogenic potential [65]. Additionally, Xu et al., coupled M2e with a cross-reactive substance (CRM197) to synthesize a novel vaccine, M2e-CRM197, which induced anti-M2e antibodies after immunization of mice, and the induction efficiency was higher than the M2e alone group, exhibiting higher immunogenicity [66]. INGROLE et al., coupled M2e to gold nanoparticles (AuNPs) and added the TLR9 agonist CpG as an adjuvant to develop a broadly protective influenza A vaccine [67]. These influenza M2e epitope-based vaccine designs show promise for the development of a universal vaccine for influenza.

### 4.4. Nucleoprotein (NP) as a Conserved Immunogen for Universal Vaccines

Nucleoprotein (NP) is the main component of the nucleocapsid and forms ribonucleoprotein complexes (RNPs) with viral genomic RNA and viral polymerase, which are involved in viral transcription, replication and assembly. NP is the internal protein of influenza virus that induces the production of non-neutralizing antibodies in the host and are also major antigens recognized by cytotoxic T lymphocytes (CTLs) [68]. CTLs exert cytotoxic effects by recognizing antigenic peptides of influenza virus NPs presented by MHC class I molecules on the surface of infected cells, thereby removing influenza virus from infected host cells [69]. Due to the highly conserved NP gene, its induced CTL response can generate cross-reactivity against homozygous and different subtypes of viral strains, inducing broader and more durable protective immunity. Therefore, influenza virus NP has also been of great interest as a next-generation alternative vaccine target. For example, Lee et al., prepared two recombinant adenoviral vector vaccines with NP to test whether variation in NP epitopes in two antigenic lineages of influenza B virus affects the dominant CTL response and protective immunity induced by vaccination and found that both vaccines induced strong cross-reactive CTL after intranasal immunization of mice, and this remained cross-protective even with CTL epitope changes, suggesting that NP is a conserved universal vaccine target [69]. In addition, a recombinant protein vaccine, termed OVX836, was obtained by fusing the DNA sequences of the heptameric structural domain of influenza A virus NP and OVX313, which Campo et al., had previously demonstrated to provide broad-spectrum protection against influenza viruses. Experiments later showed that, compared to mutant NP (NPm) and wild-type NP (NPwt), which form monomeric and trimeric structures, OVX836 immunologically induced higher numbers of CD8+ T cells with NP-specific IFN-γ-mediated protection against different influenza subtypes and is now in clinical phase I trials [70,71]. It has exhibited high immunogenicity and safety, further demonstrating that NP is a viable candidate target for a universal vaccine.

## 5. Design of Broad-Spectrum Influenza Vaccines

The basic requirement of the current universal influenza vaccine is to have ≥75% protection against influenza symptoms caused by influenza A and B viruses, and the vaccine should be effective for at least one year in all age groups [72]. HA, NA, M2e and NP have different advantages as vaccine immunogens and can induce complementary immune responses. With the intensive study of influenza virus antigens, several strategies for vaccine design have been presented. 

### 5.1. Tandem Protein Immunization

A fundamental approach to broad-spectrum influenza vaccine design is to express different conserved epitopes of viral proteins in tandem or fusion, which not only improves vaccine efficacy but also reduces manufacturing time and cost compared to vaccines targeting a single antigenic peptide of the virus. Some relevant studies are available; for example, Golchin et al., designed and synthesized the tandem antigenic protein M2e-HA2, which significantly induced a humoral immune response to influenza virus after immunization of mice, and the viral challenge proved that the tandem protein protected mice from H9N2 influenza virus [73]. Farahmand et al., designed a tandem protein 3M2e-HSP consisting of three tandem repeats of M2e amino acid sequences fused to the Leishmania protozoan HSP70 gene, and preliminary in vivo experiments in mice showed that the 3M2e-HSP chimeric protein stimulated a specific immune response and induced lymphocyte proliferation and IFN-γ secretion, protecting mice from lethal influenza challenges [74]. Kwak et al., produced a 3M2e-3HA2-NP fusion protein that was fully protective against H1N1 virus in immunized mice, inducing strong virus-specific antibody responses, cytotoxic T-cell activity and antibody-dependent cellular activity, and mice exhibited higher cross-protection when adjuvanted with polygamma-glutamic acid and alum (PGA/alum) complex [75]; these results suggest that tandem protein immunization is a feasible approach for broad-spectrum influenza vaccine design. However, the vaccine is currently poorly immunogenic, induces a weak immune response, usually requires multiple vaccinations to achieve protective immunity and has a relatively slow production rate [76].

### 5.2. Nanoparticle Presentation

The use of nanoplatforms to deliver relevant antigens is a promising approach for the development of novel influenza vaccines. The high density and structural ordering of antigens in nanoparticles promotes antigen recognition by B-cell receptors and triggers an effective cellular and humoral immune response. In addition, nanoparticles have been shown to protect antigens from proteolytic degradation, to improve antigen delivery and to prolong antigen presentation by antigen presenting cells (APCs). For example, Wei et al., constructed a biomimetic influenza vaccine using an apoferritin (AFt) nanocage as a carrier with an internally wrapped NP protein and a surface-coupled HA protein, which produced HA- and NP-specific antibodies with complete protection against lethal infection by homologous and heterologous viruses after immunization of mice, and this biomimetic vaccine suggested a vaccine development strategy [77]. Ma et al., constructed bilayer protein nanoparticles with an HA or M2e shell and an NP core and used them to immunize mice and found that the vaccine significantly increased M2e-specific serum antibody potency and enhanced concomitant ADCC responses, induced stronger NP-specific T-cell responses and HA neutralization and protected mice, not only from homologous viruses but also against heterologous and heterosubtypic influenza viruses [78]. Qiao et al. fused A-helix (Ah) and CD-helix (CDh) from the HA stem of H3N2 virus with ferritin alone or in tandem to produce Ah-f, CDh-f and (A + CD)h-f nanoparticles. After three subcutaneous immunizations of mice, CDh-f and (A + CD)h-f were found to induce a robust humoral and cellular immune response with complete protection against lethal attack by H3N2 virus [79]. Boyoglu-Barnum et al., constructed a two-component nanoparticle immunogen by computational design, displaying 20 HA trimers in an ordered array, and in vitro assembly was able to precisely control the co-display of multiple different HAs in a defined ratio. For example, when used to display four influenza virus subtypes of HA of the marketed quadrivalent influenza vaccine (QIV), triggering a vaccine-matched strain with an antibody response equivalent or superior to that of commercial QIV while also inducing a broad protective antibody response against heterologous viruses by targeting HA stems [80], these results demonstrate that nanoparticle-based platforms for antigen presentation are promising vaccine platforms against influenza. The drawback is that the size of nanoparticles cannot be accurately controlled [81]. Some data show that the larger the nanoparticles can be more effectively recognized by APC and activate T cells more strongly, but also have good lymphatic circulation [82]. In addition, the choice of antigens and the distance between antigens should be considered: different antigens bring different immunogenicity and are an important choice for vaccine products. The number of antigens and their density are also closely related to the effectiveness of the vaccine. Available data show that more antigens and more dense antigens can activate B cells more strongly, but at the same time, excessive density can affect the effectiveness of antigen binding; so, it is necessary to find the best balance between quantity and distance [83].

### 5.3. Virus-like Particles (VLPs) Combination Vaccine

VLPs are produced by inserting genes encoding viral structural proteins into plasmid vectors and then transforming or transfecting them into the target expression system. Since the produced VLPs do not contain viral nucleic acids, they are not infectious nor pathogenic, have strong immunogenicity and can stimulate strong humoral and cellular immune responses in the body. Insect cell-baculovirus expression systems are often chosen to produce influenza VLPs, and the resulting VLPs are morphologically similar to natural influenza viruses. For example, Zhang et al., prepared VLPs containing H3N8 equine influenza virus (EIV) HA, NA and M1 proteins by an insect cell-baculovirus expression system, and the hyperimmune serum produced by immunizing horses with the VLPs was injected into mice that were then challenged, showing that the equine hyperimmune serum had good prophylactic and therapeutic effects against H3N8 EIV infection [84]. Li and Hu et al., constructed an H7N9 VLP vaccine candidate using an insect cell-baculovirus expression system that contained HA, NA and M1 and elicited a strong antibody immune response after intramuscular immunization, protecting chickens and mice from lethal H7N9 virus attack, while the antibodies induced by these H7N9 VLPs exhibited good cross-reactivity [85,86]. Keshavarz et al., prepared H1N1 VLPs (containing H1, N1 and M1 antigens) by the insect cell-baculovirus expression system and injected them into mice by the intramuscular (IM) or intranasal (IN) route, resulting in strong T-cell immunity [87]. These experimental results highlight the great potential and feasibility of a VLP vaccine based on the insect cell-baculovirus expression system. However, there are limitations, i.e., baculovirus particles and VLPs are difficult to separate, and there are challenges in mass production of VLPs vaccines using this system [88]. In addition, Wei et al., proposed a strategy for influenza vaccine design based on hepatitis B virus core virus-like particles (HBc VLP) by genetic fusion to display M2e antigen on the outer surface of HBc VLP and NP antigen wrapped around the inner surface, constructing a biomimetic double-antigen influenza vaccine with internal NP/external M2e. After intraperitoneal immunization of mice, the vaccine induced the production of NP- and M2e-specific antibodies and completely protected mice from lethal H1N1 influenza virus without significant body weight loss, whereas the nonbiomimetic double-antigen vaccine provided only 62.5% protection and 12.5% body weight loss, a result that suggests that the HBc VLP-based influenza vaccine design strategy also has potential for future development [89]. Although current studies have confirmed that VLPs can be used as an effective vaccine against influenza, there are still some technical difficulties in their production, such as the assembly of influenza VLPs generally requiring several proteins, the probability of recombinant vectors expressing different proteins entering the same cell is small, which reduces the chance of interaction between proteins and VLPs assembly, and the difficulty of constructing multi-gene co-expression vectors, and the ratio of expression of each gene on the same vector is not easy to control. In addition, the formation of VLPs may be affected by the difficulty of constructing multi-gene co-expression vectors and the difficulty of controlling the expression ratio of each gene in the same vector, and the challenges of how to effectively scale up the process, improve the assembly efficiency of VLPs and reduce the cost [90,91,92].

### 5.4. DNA Vaccine

A DNA vaccine is a vaccine that uses recombinant technology to clone target genes into eukaryotic expression vectors and introduce them into the body to express the corresponding antigens and induce a protective immune response for prevention and treatment. Existing DNA-based antigen delivery platforms include bacterial plasmids, recombinant viral vectors and bacterial vectors. While conventional influenza vaccines tend to induce organism-dependent immunity and lack T-cell immunity, DNA vaccines developed with conserved epitopes of influenza virus internal proteins tend to induce T-cell responses and play an important role in inhibiting virus transmission and clearing infected cells, facilitating broader cross-protection. Moreover, recombinant DNA molecules can encode multiple target gene fragments at the same time and highly express the target protein at the cellular level, which cannot only reduce the production cost of vaccines to a certain extent but can also stimulate both humoral and cellular immune responses. However, the immunogenicity and protective potency of DNA vaccines are low, which requires the design of optimized antigenic sequences or exogenous modifications to ameliorate them. For example, Guilfoyle et al. designed a polyvalent influenza DNA vaccine encoding HA and NA from H1N1 (2009) and H3N2 (1968) strains and NP, M1 and M2 from the H1N1 (1918) strain. Immunization of ferrets with this DNA vaccine not only protected them from homologous attack by H1N1 pdm09 but also from the heterologous influenza virus H5N1, demonstrating the efficacy of this DNA vaccine against heterologous viruses [93]. Using M2e as the target antigen, Yao et al., constructed two DNA vaccines, p-tPA-p3M2e with increased antigen epitope density and p-p3M2e with enhanced antigen secretion. The results showed that both vaccines induced high M2e-specific humoral and cellular immune responses in mice, with p-tPA-p3M2e providing more effective protection against lethal attack by the same subtype of H1N1 virus and exhibiting cross-protection against different subtypes (H9N2, H6N6 and H10N8) [94]. In addition, Qiao et al., designed a DNA vaccine called HA-F by fusing influenza virus HA with self-assembled ferritin nanoparticles, and after immunizing mice with 100 µg of this DNA vaccine three times, it showed significant HA-specific humoral and T-cell immune responses and protected mice from lethal infection by homologous H1N1 virus, indicating that HA-F is a competitive vaccine candidate [95]. In summary, DNA vaccines hold promise for development as universal, broad-spectrum influenza virus vaccines. Currently, the primary mode of administration of DNA vaccines is still intramuscular, which makes it difficult for the vaccine to cross cell membranes, with only a small amount reaching antigen-presenting cells and eliciting an immune response. In addition, although good immune protection has been achieved in animal models, such as mice, the reduced efficacy often observed in large animals and humans has hindered progress in clinical applications, and the large-scale use of prophylactic DNA vaccines has raised a number of safety concerns, perhaps most notably integration into the host genome, antibiotic resistance and the potential for DNA-targeted autoimmunity [96,97].

### 5.5. mRNA Vaccine

mRNA vaccines are produced by introducing mRNA containing encoded antigenic proteins into the body, which are transcribed and translated to produce the corresponding antigenic proteins and induce a specific immune response [98]. For example, Zhuang et al., constructed an H1N1-HA mRNA vaccine and immunized mice by intranasal administration and showed that the mRNA vaccine elicited humoral and cellular immune responses and protected mice from lethal influenza viruses [99]. Freyn et al., designed an mRNA encoding multiple antigenic proteins (HA stem, NA, M2e and NP) simultaneously and immunized mice subcutaneously with nucleoside modifications and LNP encapsulation, resulting in a broad and strong immune response. Serum antibodies obtained after a single immunization with this vaccine bind multiple influenza virus strains, demonstrating the effectiveness of nucleoside-modified mRNA-LNP vaccines expressing multiple conserved antigens as universal influenza virus vaccine candidates [100]. Chivukula et al., demonstrated the suitability of mRNA vaccines using multiple influenza antigens as monovalent or polyvalent vaccines with HA and NA mRNA-LNP preparations inducing strong functional antibodies and cellular responses in nonhuman primates. Furthermore, they demonstrated that co-encapsulated and combined polyvalent vaccines with HA/NA mRNA-LNPs could effectively deliver all antigens without immune interference and that all antigens were immunogenic in the formulation when delivered alone, supporting the use of mRNA polyvalent vaccines for the production of rapid pandemic or seasonal influenza virus vaccines [101]. In vitro synthetic mRNA vaccines for several IAV subtypes have been tested in mice, ferrets and pigs, and these vaccines have been shown to be immunogenic and safe. Nonhuman primates have responded well to lipid nanoparticle-delivered mRNA vaccines, showing a sustained humoral immune response. In the last decade, mRNA vaccine production and delivery technologies have improved significantly, and it has become possible to design and develop a universal influenza mRNA vaccine [102]. However, mRNA is extremely unstable and susceptible to degradation in the environment, and mRNA vaccines need to be stored at −80 °C, which makes maintaining the cold chain extremely difficult. Safer, more effective and cold-chain-free mRNA vaccines require the development of additional mRNA delivery vectors. One example is the development of heat-resistant mRNA vaccines [103]. Further understanding of the mechanism of action of mRNA vaccines in vivo remains to be investigated, especially in the area of studying the effects of the respective mRNA, delivery system and specific target cells or organs on the innate immune response to minimize potential side effects [104,105].

### 5.6. Subunit Vaccine

Subunit vaccines are further purified antigenic components based on cleaved virus vaccines, which have a higher safety profile but are less immunogenic. Improvement of immunogenicity is often achieved by adding suitable adjuvants, fusing multiple viral proteins and changing the vaccination strategy. For example, Saleh et al., evaluated the immunogenicity and protective properties of an influenza A virus H1N1 chimeric subunit vaccine (3M2e-HA2-NP) in a mouse model and found that mice vaccinated with this chimeric subunit vaccine produced specific antibody responses and stimulated cytokines for the production of memory CD4 cells and Th1 and Th2 cells, showing 75% protection against lethal attack by influenza virus [106]. By fusing the highly conserved antigenic region of influenza virus H7N9 subtype HA protein and M2e protein with CTxB (B subunit of the cholera toxin) as an adjuvant, Jafari et al., formed an M2e-Linker-CTxB-Linker-HA2 chimeric vaccine, and bioinformatics analysis showed that the chimeric vaccine retained the viral antigenic structure and antigenicity with high stability and a long half-life [107]. Sharma et al., designed a universal influenza subunit vaccine based on the highly conserved epitope sequences of influenza viruses HA, NP and M1, which was designed to include 2 peptide adjuvants, 26 CTL epitopes, 9 HTL epitopes and 7 linear BCL epitopes to induce innate, cellular and humoral immune responses against influenza A virus. The subunit vaccine is predicted to have a stable structure, and molecular dynamics simulations suggest that the immunogenicity of the vaccine provides protection against multiple influenza A virus subtypes [108]. The above results strongly suggest that the subunit vaccine may continue to be developed as a promising influenza virus vaccine candidate. The disadvantage of this vaccine is its general inability to induce effective cell-mediated responses, particularly those mediated by CD8+ T cells, the presence of which has been shown to be associated with a better prognosis after influenza virus infection. These responses play an important role in significantly reducing viral load and improving disease severity.

Overall, among the above vaccine development strategies, mRNA vaccines are the most promising for the following reasons: First, mRNA vaccines have a short development cycle and can be rapidly developed based on the genetic sequence of the virus. Second, mRNA vaccines are a dual mechanism of action for humoral and T-cell immunity. While conventional vaccines in the past focused on the induction of humoral immunity, mRNA vaccines produce both humoral and cellular immunity, stimulating T cells to prevent virus invasion and proliferation. Third, mRNA vaccines are immunogenic and do not require the adjuvant effect, avoiding possible immune side effects caused by adjuvants. Fourthly, mRNA vaccines do not carry viral components, have no risk of infection and are degraded through normal cellular processes, making them safe. As a final point, the mRNA vaccine technology has a simple production process, is easy to mass produce, and supports the goal of global supply [109,110].

## 6. Conclusions

Influenza continues to pose a significant threat to public health, and vaccination is currently the most effective measure to combat it. Although there are currently several manufacturers of influenza vaccines in China, the product composition is homogeneous and mainly consists of lysis vaccines, which are produced in chicken embryos. In addition to the long production time, another disadvantage is that adaptive mutations may develop and thus alter the protective effect of influenza vaccines. MDCK cell-based influenza vaccines could prevent this problem although these cell line-based vaccines are currently in their infancy in China. In addition, vaccination rates in China are low compared to those in developed countries. Many factors influence vaccination rates, including vaccination policies, vaccination history and knowledge and attitudes toward influenza and vaccination. To address these factors, measures need to be proposed to improve vaccination coverage.

In addition, considering the ongoing global COVID-19 pandemic, occurrence of another influenza epidemic will generate a large number of cases with similar respiratory infection symptoms, thus increasing the complexity of the COVID-19 differential diagnosis. This can lead to the possibility that some COVID-19 cases may be difficult to detect and manage in a timely manner, which may exacerbate the risk of transmission of SARS-CoV-2 and bring about more serious health effects.

To better control seasonal influenza and possible future influenza pandemics, it is urgent to design and develop a universal influenza vaccine that can address all types and subtypes of influenza viruses and produce durable immunity. In recent years, many important advances have been made in the development of a universal influenza vaccine, and several broad-spectrum influenza vaccines based on different targets and forms have been successfully validated in animal models. Some vaccine candidates have entered the clinical phase and have demonstrated good immunogenicity and safety, but no universal influenza vaccine has been approved for marketing, which may be related to the diversity and continuous variation of influenza virus types and the ultimate efficacy of the vaccine. This may impact sustained protection, as well as the issues of quality monitoring, market regulation and relevant policies.

## Figures and Tables

**Figure 1 viruses-15-00116-f001:**
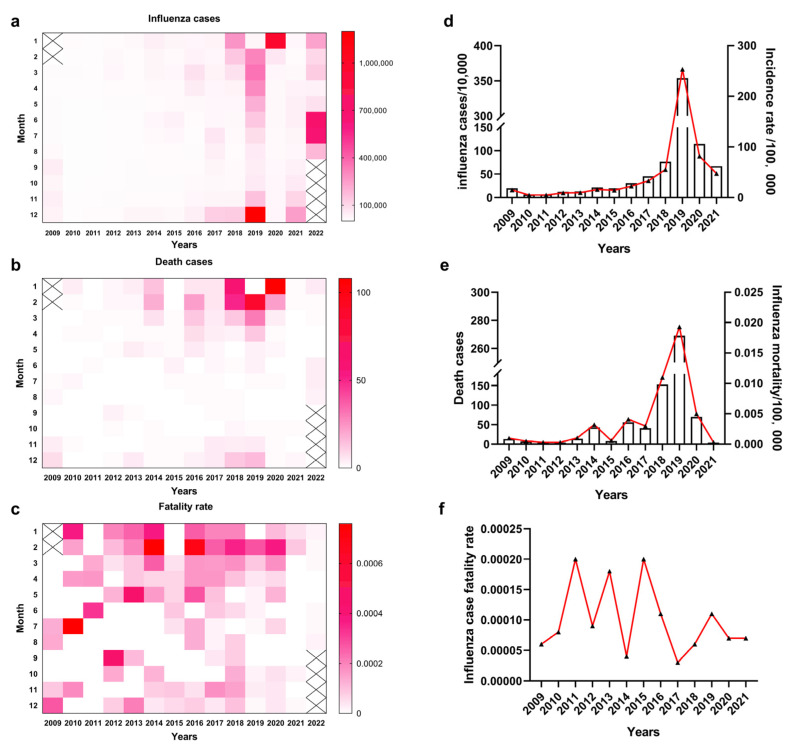
Analysis of monthly influenza infections from March 2009 to August 2022 and annual influenza infections from 2009 to 2021. (**a**) Monthly influenza infection from March 2009 to August 2022. (**b**) Monthly influenza deaths from March 2009 to August 2022. (**c**) Monthly influenza case fatality rate from March 2009 to August 2022. (**d**) Annual influenza infection and incidence rate from 2009 to 2021. (**e**) Annual influenza deaths and mortality from 2009 to 2021. (**f**) Annual influenza case fatality rate from 2009 to 2021.

**Figure 2 viruses-15-00116-f002:**
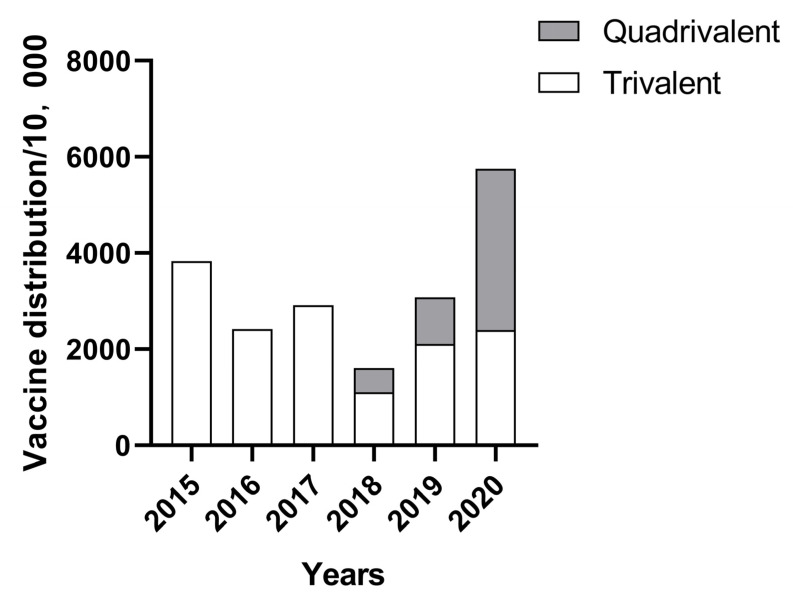
China’s influenza vaccine industry batch issue volume in recent years.

**Figure 3 viruses-15-00116-f003:**
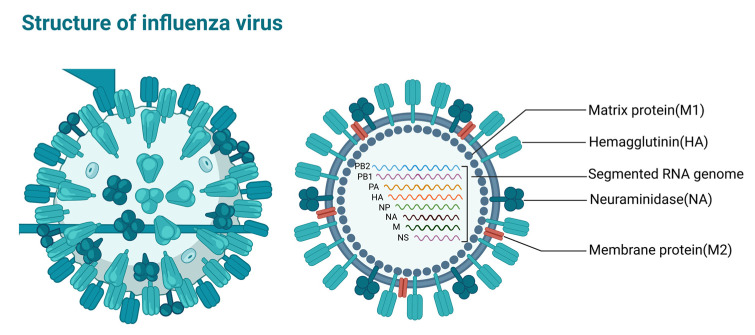
Structure of influenza virus. Image created with BioRender (https://biorender.com/ (accessed on 27 November 2022)).

**Figure 4 viruses-15-00116-f004:**
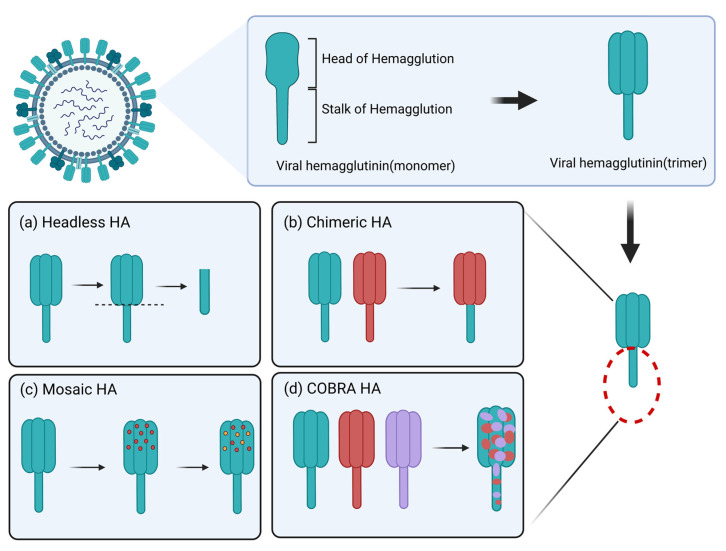
Each HA monomer consists of two parts, a spherical head and a stalk stem, and the three HA monomers are combined in a noncovalent bond to form a homotrimer. There are several strategies for the development of HA-based influenza vaccines as follows. (**a**) Complete removal of the HA head region to create a “headless” HA structure, retaining only the conserved HA stem. (**b**) Sequential immunization with chimeric HA vaccines generated using identical HA stalk regions and different HA heads that induced antibody responses are concentrated in the conserved HA stems. (**c**) Replacement of immunodominant antigenic sites in the head with sequences of other HA subtypes. (**d**) COBRA HA vaccines are developed through a series of HA protein comparisons and subsequent generation of shared sequences to optimize HA peptides that elicit a broadly reactive immune response against influenza viruses. Image created with BioRender (https://biorender.com/ (accessed on 27 November 2022)).

**Table 1 viruses-15-00116-t001:** Influenza vaccine approved and issued in China.

	Manufacture	Type of Vaccine	Inoculation Method	Specifications	Doses of Vaccination	Applicable People
Trivalent influenza vaccine	SINOVAC BIOTECH CO., LTD.	Inactivated split	Intramuscular	0.25 mL	2	6~35 months
				0.5 mL	1	≥3 years old
	Changchun Institute Of Biological Products Co., Ltd	Inactivated split	Intramuscular	0.25 ml	2	6~35 months
	Aleph Biomedical Company Limited	Inactivated split	Intramuscular	0.25 mL	2	6~35 months
				0.5 mL	1	≥3 years old
	HUALAN BIOLOGICAL ENGINEERING, INC.	Inactivated split	Intramuscular	0.5 mL	1	≥3 years old
	Shenzhen Sanofi Pasteur Biological Products Co., Ltd.	Inactivated split	Intramuscular	0.25 mL	2	6~35 months
				0.5 mL	1	≥3 years old
	Zhongyianke Biotech.Co., Ltd.	Subunit	Intramuscular	0.5 mL	1	≥3 years old
	Changchun Bcht Biotechnology Co.	Live attenuated	Nasal spray	0.2 mL	1	3~17 years old
Quadrivalent influenza vaccine	SINOVAC BIOTECH CO., LTD.	Inactivated split	Intramuscular	0.5 mL	1	≥3 years old
	Changchun Institute Of Biological Products Co., Ltd.	Inactivated split	Intramuscular	0.5 mL	1	≥3 years old
	ADIMMUNE Corporation	Inactivated split	Intramuscular	0.5 mL	1	≥3 years old
	HUALAN BIOLOGICAL ENGINEERING, INC.	Inactivated split	Intramuscular	0.25 mL	2	6~35 months
				0.5 mL	1	≥3 years old
	Jiangsu GDK Biotechnology Co., LTD.	Inactivated split	Intramuscular	0.5 mL	1	≥3 years old
	Shanghai Institute of Biological Products Co., Ltd.	Inactivated split	Intramuscular	0.5 mL	1	≥3 years old
	WUHAN INSTITUTE OF BIOLOGICAL PRODUCTS CO., LTD.	Inactivated split	Intramuscular	0.5 mL	1	≥3 years old

Note: Data from China Institute for Food and Drug Control.

## Data Availability

Not applicable.

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
