# Peer review of "Influenza and Universal Vaccine Research in China"

_viruses, 2022, doi:10.3390/v15010116_

Round 1
Reviewer 1 Report
Manuscript:
I found this an interesting manuscript presents a research topic of increasing interest entitled "Influenza and Universal Vaccine Research in China." This manuscript provided cases related to influenza virus infection and death in China in recent years, this paper describes the existing marketed vaccines, vaccine distribution and vaccination in China and summarizes the candidate immunogens designed based on the structure of influenza virus, hoping to provide ideas for the design and development of new influenza vaccines in the future. However, I note a number of places where the manuscript could be strengthened to help make it more convincing.
Major revision:
Why did the review focused on China? It can be globally, and why did the review focus on human influenza? Not poultry influenza vaccines and what is the relationship? You can add subtitles related.
Minor revision:
-Introduction: The scientific background of the topic is poor. In "Introduction" and "titles", the authors should cite recent references from JCR journals (with impact factor) about recent achievements on enhanced infection and immunity, induced stress and the genetic and antigenic analysis of avian influenza and its control. For example, authors can cite to:
1. Abdelfattah H. Eladl, Verginia M. Farag, Reham A. El‑Shafei, Abeer E. Aziza, Walaa F. Awadin and Nagah Arafat (2022): Immunological, biochemical and pathological effects of vitamin C and Arabic gum co‑administration on H9N2 avian influenza virus vaccinated and challenged laying Japanese quails. BMC Veterinary Research 18:408.
2. Mahmoud Ibrahim, Abdelfattah H. Eladl, Hesham A. Sultan, Abdel Satar Arafa, Alaa G. Abdel Razik, Sahar Abd El Rahman, Kamel I. Abou El-Azm , Yehia M. Saif, Chang-Won Lee (2013): Antigenic analysis of H5N1 highly pathogenic avian influenza viruses circulating in Egypt (2006–2012). Veterinary Microbiology 167: 651–661.
3. Ahmed Ali, Mahmoud Ibrahim, Abdelfattah H. Eladl, Yehia M. Saif, Chang-Won Lee (2013): Enhanced replication of swine influenza viruses in dexamethasone-treated Juvenile and layer turkeys. Veterinary Microbiology 162: 353–359.
Please clarify figure and conclusion.
Author Response
Thank you very much for your time involved in reviewing the manuscript. We also appreciate your clear and detailed feedback and hope that the explanation has fully addressed all of your concerns. In the remainder of this letter, we discuss each of your comments individually along with our corresponding responses.

Reviewer 2 Report
Please see the attached reviewer report.

Author Response

(The authors gave the same response as above.)

Reviewer 3 Report
The draft of review article titled "Influenza and Universal Vaccine Research in China" (viruses-2102652) by Jiali et al is nice piece of work. The draft is well written and deserve its publication in journal. Overall draft looks OK but I think author should address following minor issue before its acceptance.
1) At some place author use yuan and at some place RMB. Better to use same currency notation.
2) Author mentioned about different vaccines available against Influenza, but also need to mention major issue like poor thermal stability, short shelf life etc. Suggested to include a reference (see link) which also discuss major problem about vaccines and ways to resolve that.
https://doi.org/10.1080/14760584.2022.2053678
3) Need minor english editing at few places.
Author Response

(The authors gave the same response as above.)

Reviewer 4 Report
In this review article, Jiali Li and colleagues focus on recent progresses in the understanding of Influenza vaccine Research, statistically analyses the epidemiologic status of influenza in China, elaborates the existing marketed vaccines, vaccine distribution and vaccination in China and summarizes the candidate immunogens designed based on the structure of influenza virus, which can help us better understand and prevent and control influenza. The topic is of high interest and the manuscript is very organized. My only comment is that though the authors summarized and introduced many influenza vaccines, and indicated all these vaccines can be developed as promising influenza virus vaccine candidates, they did not compare the advantages and disadvantages of these vaccines and discuss which one is the most promising influenza vaccine. These can be supplemented.
Author Response

(The authors gave the same response as above.)
